# RADIO MAP PREDICTION VIA NEURAL NETWORKS WITH GROUND TRUTH SHORTCUTS AND SELECTIVE SAMPLING

*Mengfan Wu, Marco Skocaj, and Mate Boban*

Munich Research Center, Huawei Technologies
{mengfan.wu, marco.skocaj, mate.boban}@huawei.com

## ABSTRACT

In this paper, we present a machine learning approach for fast indoor radio map generation assisted by sample measurements in the target environment as extra input. Our solution is developed for MLSP 2025 The Sampling-Assisted Pathloss Radio Map Prediction Data Competition. In addition to feature engineering for input augmentation, we design a shortcut path in the convolutional neural network that routes the sample input channel directly to the deeper layers, which facilitates efficient refinement of the output radio map. We further propose selective sampling strategies for measurement locations to enhance the accuracy of the generated radio maps. The proposed method demonstrates particularly strong performance under conditions of relatively high sampling rates.

*Index Terms*— Wireless communications, radio maps, machine learning, computer vision

## 1. INTRODUCTION

Radio maps, in the form of an overall signal strength for each location in a given environment, can serve as a powerful tool for designing and optimizing wireless communication systems. Ray tracing techniques have been the foundation of radio map generation, providing versatility in environment simulation as well as accurate results. However, due to their intense demand for computational resources, generating radio map via ray tracing suffers from significant computational delay. Additionally, ray tracers require precise modeling of propagation environment (e.g., material characteristics) to produce reliable outputs, and they are unable to model (even slightly different) unseen environments. These limitations hinders their applicability in real systems. With the rise of graphics-processing-unit (GPU) based ray tracing tools (such as, e.g., Sionna [1]), the aforementioned challenge in terms of delay is tackled through increased computational capabilities, while the stringent modeling requirements and limited applicability remain.

With the recent advances in the field of computer vision and machine learning (ML), neural networks (NNs) have been explored for radio map generation thanks to mature image-based techniques and readily available resources for developing and training models. ML models, especially those based on NNs, capable of modeling complex nonlinearities between the input and target output, serve as a suitable tool for capturing the intrinsics of electromagnetic propagation. With optimized GPU-toolkit, the deployment of a NN is highly parallelized and fast. The aforementioned advantages of NNs showcase a promising frontier in radio map generation techniques.

Following the breakthrough work of [2], where a UNet [3] model is applied for outdoor radio map generation, the ML techniques have been further refined [4] and utilized in a wider scope of scenarios, e.g., indoor radio map generation [5]. Compared to outdoor propagation environment, indoor wireless propagation is typically more challenging due to the increased dominance of diffraction and reflection [6].

To this end, the First Indoor Pathloss Radio Map Prediction Challenge [7] was organized and various approaches have been proposed by the participants to address the issues. Our participation in the previous challenge resulted in a fruitful outcome. However, the challenges in modeling diffraction and reflection remain in place, as we observe the major erroneous predictions in the radio map are located in the region where reflections and diffractions dominate. To tackle the remaining issues, the Sampling-Assisted Pathloss Radio Map Prediction Data Competition [8] is proposed, which aims to combine the existing techniques in ML-based radio map with realistic but limited support from ray tracing or measurements. The residual input, provided as ground truth samples, aims to guide the learning process toward correcting regions with high prediction errors, often caused by effects such as reflection and diffraction.

In this paper, we propose a neural network architecture tailored to the integration and processing of additional ground truth samples. We further improve the feature engineering algorithms proposed in our previous work [9] by reducing computational complexity. We also derive selective sampling methods according to different sampling rates. These strate-

This work has been partially performed in the framework of the HORIZONJU-SNS-2022 project TIMES, cofunded by the European Union. Views and opinions expressed are however those of the author(s) only and do not necessarily reflect those of the European Union.

gies aim to make the most efficient use of the costly and time-consuming ray tracing or measurement process by prioritizing sample locations that yield the greatest predictive benefit.

Our solution provides significant improvement in terms of radio map accuracy (around $43\%$ reduction in errors under high sampling rate), consequently ranking sixth in the challenge.

## 2. CHALLENGE DESCRIPTION

The MLSP 2025 Sampling-Assisted Pathloss Radio Map Prediction Challenge [8] comprises two tasks, both targeting the prediction of radio maps simulated at $868$ MHz with an omni-directional antenna. The dataset, which is a subset of the previous challenge [10], includes training data from 25 buildings, each with 50 transmitter locations. To support generalization, radio maps at $1.8$ GHz and $3.5$ GHz are also deployed for training. The two tasks differ in how sample measurements can be utilized: in Task 1, sampling locations for both $0.5\%$ and $0.02\%$ sampling rates are randomly generated, whereas in Task 2, participants are free to select the sampling locations under the same rate constraints.

## 3. PROPOSED SOLUTION

This section details our solution and is organized into: feature engineering, ML model structure, sampling location selection, concluding with specifics of the training process.

### 3.1. Data Processing

Based on the provided features (reflectance, transmittance, distance to the transmitter (Tx), as shown in Figure 1), we derive two extra features as direct guidance to wireless propagation characteristics.

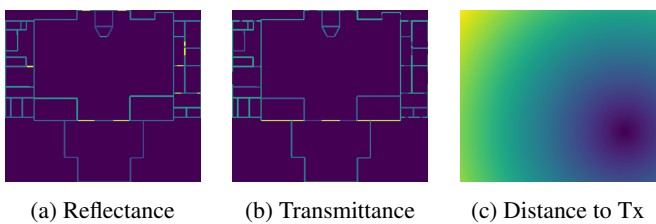

(a) Reflectance     (b) Transmittance     (c) Distance to Tx

**Fig. 1**: Original features with color representing pixels' numerical values.

#### 3.1.1. Ray-Marching Feature

The original features of reflectance and transmittance visually represent the indoor environment, with non-zero pixels corresponding to the walls/objects. From the numerical perspective, the inputs features of reflectance and transmittance

are sparse matrices while the target pathloss is dense and continuous. To this end, in our previous work [9], we derived an extra feature based on the sparse feature channel of transmittance but match the characteristics of continuity in the desired output. For clarity and also providing improvement on the previous solution, we report the corresponding procedures here as well.

Put into the context of radio propagation, for each pixel, our algorithm casts a ray from the transmitter's location and accumulates the transmittance values of the walls between the destination and source, which can be interpreted as an initial estimate of pathloss considering attenuation due to material absorption only.

We name the computed ray-marching feature based on transmittance as transmittance-spread, which can be expressed as:

$$T_{\text{spread}}(x,y) = \sum_{p \in \mathbb{L}(x_0, y_0 \to (x,y))} t(p),$$

where $t(\cdot)$ denotes the value of transmittance feature at a location, and $\mathbb{L}(x_0, y_0 \to (x,y))$ is a line formed by pixels computed by the Bresenham's line algorithm [11]. The results of this preprocessing operation is visually depicted in Figure 2a. In practice, only the pixels on the edge of image are selected as the target location and all the pixels along the ray from the source is filled with the computed values sequentially. To manage pixel overlaps between Bresenham's lines computed for adjacent edge targets, each new line begins filling pixels from the first point where it diverges from the previously drawn line.

Our algorithm achieves single-thread runtime of less than 100 ms. With parallelization of the algorithm on different sample inputs, an average run time of 7 ms is achieved.

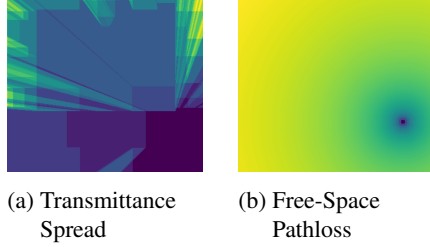

(a) Transmittance     (b) Free-Space
Spread            Pathloss

**Fig. 2**: Derived features with color representing pixels' numerical values

#### 3.1.2. Free-Space Pathloss

Another estimation of the pathloss considering free-space propagation only is the large-scale pathloss. Expressed in decibels, the feature is computed as:

$$L_{fs} = 20 \cdot \log_{10}\left(\frac{4\pi df}{c}\right),$$

where $d$ is the propagation distance, $c$ is the speed of light, and $f$ is the frequency of the transmitter's signal. A sample free-space pathloss feature is shown in Figure 2b.

## 3.2. Sampling Measurements

For each task in the challenge, pathloss values are provided for a subset of locations, which can be used to enhance prediction accuracy. In our solution, we incorporate an additional input channel where sampled pathloss values are placed at their corresponding pixel locations, with all other pixels set to zero. This channel is used during both training and prediction. We illustrate the target pathloss, $0.5\%$-sampled pathloss, and $0.02\%$-sampled pathloss in Figure 3

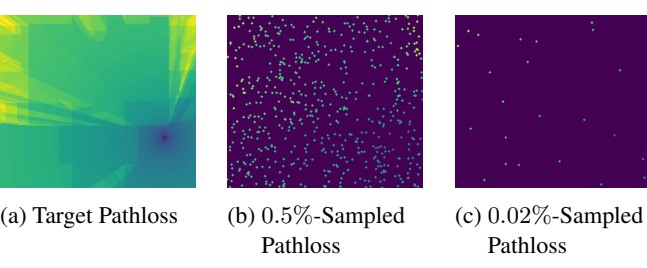

(a) Target Pathloss  (b) 0.5%-Sampled Pathloss  (c) 0.02%-Sampled Pathloss

**Fig. 3**: Target pathloss and sampled pathloss with color representing pixels' numerical values.

## 3.3. Data Augmentation

All features, including the original, the derived, and the sample channel are resized to $512 \times 512$ pixels with bilinear interpolation. Moreover, random flipping and rotation of the inputs are performed during training.

## 3.4. Model Structure

As the radio map prediction conforms to an image-to-image translation problem, we employ a U-Net based model for the task at hand. Unlike in our previous work [9], where sparse features and dense features are processed through two parallel encoder-decoder streams, we opt here for a simpler processing based on a single up/down convolutional stream. The motivation behind this design is to not only reduce training and inference time, but also enable the sample channel to guide the correction of features related to free-space pathloss and transmittance spread. Before each upsampling and downsampling operation in the U-Net architecture, we apply a double convolution with residual connection, where both of the blocks or the second block use stacked dilated convolution to effectively capture and process sparse features.

Specifics of the stacked dilated convolution are shown in Table 1, with the number of convolution units (channels) decrease as the dilation rate increases.

**Table 1**: Components of stacked dilated convolution

| Name | Kernel Size | Dilation | Channel |
|------|-------------|----------|---------|
| r1   | $3 \times 3$ | 1  | $n/2$  |
| r3   | $3 \times 3$ | 3  | $n/4$  |
| r6   | $3 \times 3$ | 6  | $n/8$  |
| r9   | $3 \times 3$ | 9  | $n/16$ |
| r12  | $3 \times 3$ | 12 | $n/16$ |

**Table 2**: Structure of customized double convolution for tasks

| Task | Sampling Rate (%) | Double Convolution | Channel Doubling Base ($n$) |
|------|-------------------|--------------------|-----------------------------|
| 1 | 0.5  | conventional +stacked dilated | 64 |
| 1 | 0.02 | stacked dilated +stacked dilated | 64 |
| 2 | 0.5  | conventional +stacked dilated | 80 |
| 2 | 0.02 | stacked dilated +stacked dilated | 64 |

To directly refine the predicted pathloss, we introduce a shortcut pathway that connects the sample input channel to the later stages of the U-Net architecture. As illustrated in Figure 4, the sample channel is processed through both the main U-Net stream and this shortcut connection, which merges with the U-Net's output. To propagate sampled pathloss information to nearby regions, we apply a transposed convolution along the shortcut path, effectively "diffusing" the values to neighboring pixels. The kernel size of the transposed convolution is set to $3 \times 3$ for a higher sampling rate of $0.5\%$, and $5 \times 5$ for a lower sampling rate of $0.02\%$. The resulting diffused sample features are concatenated with the U-Net output and passed through two additional standard convolutional layers to produce the final prediction.

In the U-Net architecture, the number of channels doubles progressively toward the bottleneck layer. For Task 2, which uses a sampling rate of $0.5\%$, we increase the initial number of channels to $80$—higher than the standard $64$ used in other tasks—to more effectively capture the additional information provided by the densely-sampled grid points. The configurations of the double convolution units employed for each task are detailed in Table 2.

## 3.5. Sampling Locations

The second task of the challenge allows participants to select sampling locations strategically to acquire pathloss values that can inform and improve prediction accuracy. For the case of high sampling rate, the sampling density on one axis is around $7\%$. With the aforementioned diffusion layer (size

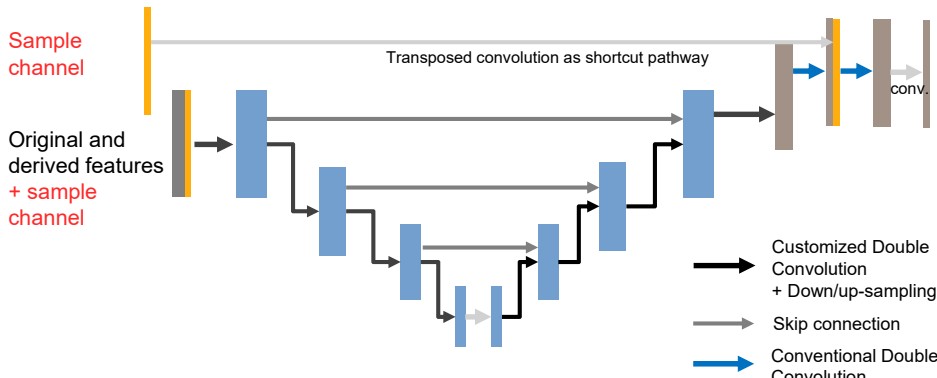

**Fig. 4**: A 4-layer U-Net with customized double convolution blocks and shortcut pathway for sample channel.

$3 \times 3$) spreading the sampled values to two neighboring pixels, it is possible to achieve a coverage of $20\%$ on each axis. As a result, a straightforward uniform grid sampling strategy – excluding points near the transmitter to comply with the sampling location limit – proves to be an effective and practical choice. Resulting sampling locations are illustrated in Figure 5a. This approach allows the generation of a uniform feature channel via transposed convolution using uniformly spaced samples, while excluding regions near the transmitter where propagation is generally less complex.

In the second task, which involves a sparse sampling rate, it is essential to strategically select sampling locations that are likely to correspond to regions with higher prediction errors. Based on insights from our earlier work [9], we suppose that areas with strong reflection effects pose particular modeling challenges. To address this in Task 2, we first apply the Hough Line Transform [12] to the reflectance feature in order to detect wall structures. For each detected wall, we identify the pixel with the highest reflectance value—referred to as the most reflective point—and estimate the wall's orientation based on its start and end coordinates. Using this information, our algorithm casts rays toward the reflector (tx_ray in Figure 5b) and selects one pixel along the estimated reflection path (rf_ray in Figure 5b) as a sample point. This procedure is repeated until the predefined number of sample points is reached.

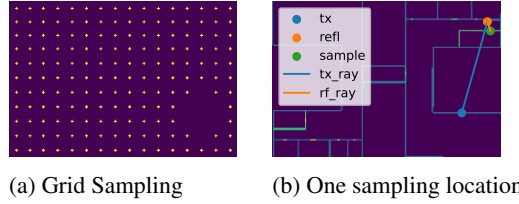

(a) Grid Sampling    (b) One sampling location based on Reflection

**Fig. 5**: Two sampling methods

**Table 3**: Official evaluation results on both tasks under different sampling rates

| Task | Sampling Rate (%) | RMSE on Training Data | Test RMSE | Inference Time (ms) |
|---|---|---|---|---|
| 1 | 0.5 | 4.00 | 4.18 | 41.37 |
| 1 | 0.02 | 4.81 | 6.91 | 54.35 |
| 2 | 0.5 | 2.95 | 3.81 | 51.37 |
| 2 | 0.02 | 4.80 | 6.91 | 55.54 |

### 3.6. Training

Models are trained separately for each task and sampling rate, using a batch size of 2 and an initial learning rate of $0.002$. The learning rate is managed with a scheduler that reduces it when the validation performance plateaus. For Task 1, early stopping is applied based on performance on a held-out test set. For Task 2, where no dedicated test set is available, we employ cross-validation to determine the point at which the model begins to overfit, as indicated by a rise in validation error.

We use a composite training loss that combines mean squared error (MSE) with the structural similarity index measure (SSIM) [13], formulated as: $L = 0.8 \cdot \mathrm{MSE} + 0.2 \cdot (1 - \mathrm{SSIM})$. This convex combination balances pixel-wise accuracy with perceptual similarity. For final evaluation, the models are typically trained for 27 epochs on the full training dataset, except for Task 2 with a $0.5\%$ sampling rate, which is trained for 23 epochs.

### 4. RESULTS AND DISCUSSIONS

We show the prediction error on a training data sample in a heatmap of the same scale in Figure 6, as well as the root mean squared errors (RMSE) in Table 3. Note that the Task 1 model for $0.02\%$ sampling rate is also directly used for Task 2 of $0.02\%$ sampling rate with reflector-based sampling.

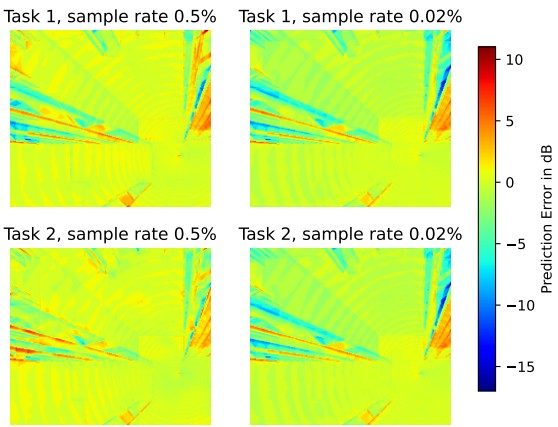

**Fig. 6**: Comparison of model prediction errors at different sampling rates for a representative training sample.

We observe that grid-based sampling in Task 2 with a higher sampling rate of $0.5\%$ leads to a significant improvement in prediction performance, achieving a $9\%$ reduction in RMSE compared with random sampling in Task 1. In contrast, our reflection-based sampling at a lower rate of $0.02\%$ has not proven effective in enhancing prediction. A likely reason for this ineffectiveness is the difficulty in reliably identifying wall locations. Our analysis shows that the current line-extraction algorithm fails for certain building layouts, either missing walls entirely or producing incorrect angle information. These inaccuracies can misguide the estimation of reflection directions, thereby degrading the performance of the model.

**Table 4**: Ablation study A: Model without shortcut pathway and without sample input

| Model Structure Corresponding to | RMSE on Training Data | Test RMSE | Inference Time (ms) |
|---|---|---|---|
| Task 1 $0.5\%$ | 4.89 | 7.31 | 38.57 |
| Task 1 $0.02\%$ | 4.88 | 6.71 | 53.32 |
| Task 2 $0.5\%$ | 4.70 | 6.75 | 49.26 |
| Task 2 $0.02\%$ | 4.88 | 6.71 | 53.32 |

To evaluate the roles of the sample pathloss inputs and the shortcut pathway, we conducted an ablation study with three sets of experiments: A. Conventional U-Net-style models without shortcut pathway and without sample pathloss input (see results in Table 4), B. Models with the shortcut pathway but trained with zero-valued sample inputs, disabling the influence of samples (see Table 5), and C. Conventional U-Net-style models without shortcut pathway but trained with sample pathloss as an additional input channel (see Table 6).

Our final competition submission (shortcut pathway with valid sample pathloss input) consistently performed best on the training data across all configurations. However, in the official evaluation, our method only outperformed the ablation variants in Task 2 with the higher sampling rate ($0.5\%$), suggesting that the shortcut-enhanced design is most effective when paired with grid-based, uniformly distributed samples. This behavior can be explained by the nature of the shortcut pathway, which directly influences the output based on the sample inputs. When the samples are uniformly distributed (as in Task 2), they systematically guide the correction of the predicted map. In contrast, randomly placed samples (as in Task 1) may lead to unreliable corrections: overlapping or clustered samples can introduce noise, while sample-sparse regions may lack sufficient guidance. This explains the slightly worse performance of our official Task 1 model compared to the ablation variant C (4.18 vs 4.15).

In terms of sampling effectiveness, we observed that at the higher sampling rate of $0.5\%$, incorporating sample pathloss information—regardless of whether a shortcut pathway was used—led to approximately a $40\%$ reduction in test RMSE. This is evident when comparing the official evaluation result to that of ablation variant B, as well as comparing variant C to A. These findings validate the effectiveness of high-rate sampling, demonstrating that the inclusion of pathloss samples can significantly enhance prediction accuracy when sufficient measurement data is available.

At the lower sampling rate ($0.02\%$), the advantages of using sparse samples reversed. Both our official solution and ablation study C performed worse than models that excluded sample input (variants A and B). This is consistent with observations from the top-ranked team [14], who noted that highly sparse and randomly located samples are not representative of the true propagation patterns, and may even degrade performance when used as direct inputs during training. The problem is further exacerbated by our shortcut design: the sparse sample "noise" is propagated directly through the shortcut and further processed by the two subsequent convolution layers, leading to stronger overfitting or misguidance. This is reflected in the official evaluation score being worse than that of ablation variant C (6.91 vs. 6.82). In contrast, when there is no disturbance from sparse sample "noise", the two extra convolution layers help the model to better exploit the other input features. This is evidenced by ablation variant B outperforming variant A (6.53 vs 6.71). These results highlight the need for further refinement of both the architecture and sampling strategy—particularly in low-sample regimes—to ensure that sparse measurements are utilized effectively rather than becoming a source of noise or overfitting.

## 5. CONCLUSIONS AND FUTURE WORK

In summary, our experiments demonstrate that integrating sample inputs and architectural enhancements can significantly boost radio map prediction accuracy, particularly at higher sampling rates. However, at extremely low sampling rates, the current approach struggles to generalize effectively,

**Table 5**: Ablation study B: Model with shortcut pathway but trained with zero-valued sample input

| Model Structure Corresponding to | RMSE on Training Data | Test RMSE | Inference Time (ms) |
|---|---|---|---|
| Task 1 0.5 % | 4.83 | 6.95 | 42.88 |
| Task 1 0.02% | 4.93 | 6.53 | 54.72 |
| Task 2 0.5 % | 4.79 | 6.86 | 50.89 |
| Task 2 0.02% | 4.93 | 6.53 | 54.72 |

**Table 6**: Ablation study C: Model without shortcut pathway but using sample pathloss input

| Task | Sampling Rate (%) | RMSE on Training Data | Test RMSE | Inference Time (ms) |
|---|---|---|---|---|
| 1 | 0.5 | 4.00 | 4.15 | 38.76 |
| 1 | 0.02 | 5.04 | 6.82 | 51.37 |
| 2 | 0.5 | 2.97 | 3.88 | 47.88 |
| 2 | 0.02 | 5.04 | 6.81 | 51.78 |

indicating limitations in both the sampling strategy and model robustness. Future work will investigate whether prediction errors correlate with strong reflection effects and refine the line-extraction algorithm to improve wall detection accuracy. Enhancements to the neural network architecture – such as more effective shortcut pathways and tailored convolutional layers – will also be explored. Furthermore, inspired by the success of other top-performing teams, future research will consider refinement-based training strategy, and also designing task-specific loss functions to better capture the complex propagation characteristics inherent in indoor environments.

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
