# OpenReview forum: "Radio Map Prediction via Neural Networks with Ground Truth Shortcuts and Selective Sampling"
_IEEE.org/MLSP/2025_SA_Radio_Map_Prediction_Challenge — SA Radio Map Prediction Challenge at MLSP 2025 Oral_

### Official Review · Reviewer_DW4u · 2025-06-05
**Reviewer Comments on “Sampling-Assisted Neural Network Radio Map Generation with Shortcut Channel and Selective Sampling”**

**Rating:** 7
**Confidence:** 3

**Review:**

This paper proposes a machine learning method for fast indoor radio map generation assisted by sample measurements in the target environment as extra input. Simulation results demonstrate that integrating sample inputs and architectural enhancements can significantly boost radio map prediction accuracy, particularly at higher sampling rates. However, several issues remain to be addressed:
1) The phrasing of the paper title appears problematic, particularly the use of “sampling-assisted neural network radio map generation”.
2) The expression “where t(·) denotes the …” should not be indented at the beginning of the paragraph.
3) The subsection “3.2.1. Data Augmentation” is incorrectly positioned; it should not be a subpart of “3.2. Sampling Measurements.”
4) There are issues with the presentation of Fig. 4. The illustration suggests an incorrect interpretation, whereas it should depict a concatenation along the channel dimension.
5) There are problems with abbreviation definitions, such as “structural similarity index (SSIM),” which is improperly introduced.
6) The formatting of figure captions is inconsistent; for instance, the caption of Fig. 6 does not match the style of other figures.
7) Section “6. REFERENCES” should not begin on a separate page.
8) In Reference [10], the hyperlink is unnecessarily repeated and should appear only once.

---

### Official Review · Reviewer_hZPQ · 2025-06-05
**Comments on “Sampling-Assisted Neural Network Radio Map Generation with Shortcut Channel and Selective Sampling"**

**Rating:** 7
**Confidence:** 4

**Review:**

This paper presents a machine learning approach for indoor radio map prediction enhanced by sparse pathloss measurements. The authors propose a modified U-Net architecture that incorporates handcrafted features, a shortcut channel for sparse sample propagation, and task-specific sampling strategies. The method is evaluated on the MLSP 2025 challenge dataset and achieves competitive performance, particularly under higher sampling rates.

The proposed contributions are practically valuable and the technical methods are mostly well justified. However, several issues should be addressed to improve clarity and presentation quality.

1. Several abbreviations are used without proper definition. For example, structural similarity index (SSIM) in Section 3.5 doesn't comply with abbreviation convetions, and Tx is shown in the caption of Fig.1(c) but never defined in the text.

2. Data Augmentation is currently under section 3.2 "Sampling Measurements'', though the described augmentation applies to all inputs. It should be lifted to a standalone sub-subsection.

3. The caption for Fig.6 is less formal than others. Consistent style across all figures should be ensured.

4. In Section 3.1.1, the line beginning with "where $t(\cdot)$ denotes ...'' is improperly indented and should be inline with the previous equation explanation.

5. The references section (Section 6) starts on a new page, which is unconventional unless explicitly required.

6. The ablation study in Table 4 is intended to isolate the effect of incorporating sample points and shortcut pathways. However, only one factor—e.g., presence or absence of sample points—should be changed at a time while keeping the architecture otherwise identical to evaluate the effect of the sample points.

7. The numerical data alignment is inconsistent across all tables—Table 1 is center-aligned while the others are left-aligned. For readability and professional presentation, numeric columns should be uniformly aligned.

---

### Official Review · Reviewer_jvoe · 2025-06-08
**Solid technical contributions with minor presentation issues**

**Rating:** 7
**Confidence:** 4

**Review:**

This paper presents a well-motivated machine learning approach for indoor radio map prediction with sparse measurements. The technical contributions are solid—the shortcut pathway design and feature engineering (ray-marching, free-space pathloss) demonstrate clear value. The experimental results are impressive, achieving 43% error reduction at higher sampling rates and ranking sixth in the challenge.
The methodology is sound and the ablation study effectively validates the proposed components. The work addresses an important practical problem with a thoughtful architectural solution.
Minor presentation issues should be addressed: SSIM abbreviation formatting ("structural similarity index (SSIM)"), Data Augmentation subsection placement (currently under Sampling Measurements rather than standalone), Figure 6 caption style inconsistency, and References section formatting. These are easily correctable and don't detract from the strong technical content.
The paper makes meaningful contributions to ML-based radio propagation modeling and demonstrates practical value for the wireless communications community.

---

### Official Review · Reviewer_U5uY · 2025-06-09
**Review for "Sampling-Assisted Neural Network Radio Map Generation with Shortcut Channel and Selective Sampling"**

**Rating:** 7
**Confidence:** 4

**Review:**

- "Compared to outdoor propagation environment, indoor wireless propagation is typically more challenging due to the increased dominance of diffraction and reflection [6]." : Could the authors please provide more information on these statements? For example, which sub-sections of the referenced paper provide this information? What about scattering?
-  Precisely which layer or layers is the sample input connected to through the shortcut pathway?
-  Are there any ablation studies on the importance of the shortcut pathway when using the samples as input?
- Table 4 caption: It would be better to repeat "without" before "Shortcut Pathway".
- Table 3 caption: "with Different Sampling Rate", "Rates" might be more accurate.
- The captions might not require capitalization.